# Clinical Application for Tissue Engineering Focused on Materials

**DOI:** 10.3390/biomedicines10061439

**Published:** 2022-06-17

**Authors:** Takahiro Kitsuka, Rikako Hama, Anudari Ulziibayar, Yuichi Matsuzaki, John Kelly, Toshiharu Shinoka

**Affiliations:** 1Center for Regenerative Medicine, Nationwide Children’s Hospital, Columbus, OH 43205, USA; takahiro.kitsuka@nationwidechildrens.org (T.K.); rikako.hama@nationwidechildrens.org (R.H.); anudari.ulziibayar@nationwidechildrens.org (A.U.); bokumatsuzaki@hotmail.co.jp (Y.M.); john.kelly@nationwidechildrens.org (J.K.); 2Department of Biotechnology and Life Science, Graduate School of Engineering, Tokyo University of Agriculture and Technology, 2-24-16 Naka-Cho, Koganei 184-8588, Japan; 3Department of Cardiothoracic Surgery, Nationwide Children’s Hospital, Columbus, OH 43205, USA; 4Department of Surgery, The Ohio State University Wexner Medical Center, Columbus, OH 43210, USA

**Keywords:** tissue engineering, tissue-engineered vascular grafts (TEVGs), biodegradable scaffolds, synthetic polymers, silk fibroin, decellularized tissue, electrospinning, 3D printing, clinical trials

## Abstract

Cardiovascular-related medical conditions remain a significant cause of death worldwide despite the advent of tissue engineering research more than half a century ago. Although autologous tissue is still the preferred treatment, donor tissue is limited, and there remains a need for tissue-engineered vascular grafts (TEVGs). The production of extensive vascular tissue (>1 cm^3^) in vitro meets the clinical needs of tissue grafts and biological research applications. The use of TEVGs in human patients remains limited due to issues related to thrombogenesis and stenosis. In addition to the advancement of simple manufacturing methods, the shift of attention to the combination of synthetic polymers and bio-derived materials and cell sources has enabled synergistic combinations of vascular tissue development. This review details the selection of biomaterials, cell sources and relevant clinical trials related to large diameter vascular grafts. Finally, we will discuss the remaining challenges in the tissue engineering field resulting from complex requirements by covering both basic and clinical research from the perspective of material design.

## 1. Introduction

The annual cardiovascular disease (CVD) mortality rate is expected to increase to 23.3 million worldwide by 2030 [1]. Thus, there is a growing demand for vascular conduits for reconstructing or bypassing vascular occlusion and aneurysms. Stenosis, or blockage of blood vessels, leads to various disorders that cause tissue damage due to insufficient nutrient supply due to decreased blood flow [2].

In 1952, Voorhees et al. first reported large-diameter human blood vessels made of synthetic materials to replace obstructed arterial blood vessels [3]. Synthetic polymers such as Teflon^®^ (expanded polytetrafluoroethylene (ePTFE)), Dacron^®^ (polyethylene terephthalate (PET)), and polyurethane (PU) are the basis of artificial blood vessels from the early days of vascular engineering to the present due to their blood compatibility [4] and conduits made of these materials are commercially available for medium to large diameter artificial blood vessels. While it serves as a long-term in vivo stability degradation and mechanical support, it is always less permeable, not suitable for the spread of appropriate nutrients, and is biologically inappropriate at the blood interface. Due to these limitations, reoperation is frequently required due to stenosis of the grafts due to thrombosis or intimal hyperplasia beyond ten years. The formulation of various biomaterials, coupled with the sowing of these synthetic grafts by vascular cells, has emerged to overcome this.

Thus, tissue-engineered vascular grafts (TEVGs) have placed continued focus on using the patient’s cells and tissues to allow the replacement of native tissue that can remodel and grow with the patient over time [5,6,7]. Biofabrication has made considerable progress in the 35 years since Weinberg and Bell combined collagen and vascular cells [8]. However, it was also shown that simply using bio-derived materials was not sufficient to reproduce the function of living blood vessels due to the lack of appropriate mechanical properties. Therefore, the next step is the study of vascular tissue engineering focused on understanding the structural components of blood vessels and increasing cell adhesion and tissue formation, mimicking structure-based functions well. We must replicate native blood vessel structures, containing endothelial cells and fibroblasts that reproduce extracellular matrices inherent in fully mature vascular tissue and smooth muscle cells essential to ensuring permeability and mechanical strength.

Therefore, in this paper, we reorganize the functions based on the structure characteristic of each blood vessel and introduce materials that have been widely used in the latest graft designs. Next, we detail the remaining challenges in vascular tissue engineering through the outcomes and challenges learned from pre-clinical evaluation in vivo and clinical application in humans. The aim is to propose potential solutions and prospects in this highly dynamic field of research. TEVGs should be as similar to natural blood vessels as possible and be able to remodel, grow, self-repair, and respond directly to the environment [9,10,11,12]. Tissue-engineered vascular grafts have to perform well from when implanted to when the neotissue is fully matured. As a result, they must meet specific requirements from when they are implanted. In particular, it is necessary to appropriately match the size the size of the vessel to which they are implanted and provide appropriate strength and blood delivery. The immediate challenges after implantation are thrombus formation and immunological rejection. Early intimal development and extracellular matrix (ECM) production reduces the risk of thrombus formation and lowers the risk of immunologic rejection. This has shifted the focus from developing synthetic polymer-based materials without cells to the concept of regenerated blood vessels using cells and ECM [13]. In the long term, neotissue should match the anatomy of the native vessel that the graft is intended to replace. Below, we discuss the anatomy of vessels, focusing on the functions endowed by that anatomy to inform design requirements for TEVGs (Figure 1).

## 2. Types and Structures of Blood Vessels

When considering functional tissue replacements, it is essential to mimic the defined characteristics of native blood vessels. Understanding and recreating native properties is key to restoring the patient’s anatomy to as close to natural conditions as possible. Since the construct is a conduit to support blood flow, it must withstand the pressure applied by this flow without experiencing mechanical failure. Additionally, lumen surface characteristics must be appropriate to avoid thrombus formation. Many of the functions of blood vessels come from different compositions at the biomolecular level, and structural components of various vascular layers are essential to reproduce vascular tissue fully. The primary function of the arteries and veins is to provide efficient transport to remote locations. On the other hand, capillaries and arteries allow an optimal exchange of nutrients, oxygen, and waste products in organs and tissues. The arteries are the largest and most rugged, and arterial strengthening/replacement is often required to restore healthy blood flow.

Thick blood vessels consist of three concentric layers, the inner membrane, the middle membrane, and the outer membrane, which are common elements. This highly coupled layer structure copes with pressure and blood volume delivery conditions, with pressure and flow varying slightly based on vascular location. The arteries and large veins with a diameter of more than 6 mm are arranged for efficient transport over long distances. The large-diameter blood vessels that are the main topic of this review are incomplete in order to prevent long-term blockage. To continue to exhibit the same mechanical properties as the connecting native blood vessels, it is necessary that self-tissue develops sufficiently and acquires structural stability. Therefore, the requirements and design approaches for designing large- and small-diameter vessels are entirely different.

To achieve this goal, artificial blood vessel scaffolds must be biodegradable and permeable to maintain good biocompatibility and sufficient mechanical properties and must be sized to suit each site of use and disease state. Various methods such as sheet rolling, direct scaffolding, matrix molding, and 3D bioprinting have been developed to produce structures that meet these requirements. These make it possible to mimic the design of large vessels required for in vitro biological experiments and in vivo applications. As tissue-engineered vascular grafts are primarily being developed for medium to large vessels, this anatomy discussion will focus on vessels of intermediate to large diameter.

### 2.1. Tunica Intima

The inner layer inner membrane is the thinnest layer. Tunica intima is formed from a single continuous layer of endothelial cells and is supported by the subepithelial layer of connective tissue and supporting cells [10]. Endothelial cells (ECs) play a vital role in biological processes such as coagulation, inflammation, barrier function, blood flow regulation, and synthesis/decomposition of ECM components. The inner membrane is surrounded by a thin membrane composed of elastic fibers parallel to the blood vessels. 

### 2.2. Tunica Media

The tunica media is the middle layer of the vessel, and consists of smooth muscle cells, elastin fibers, and connective tissue arranged in concentric layers. This region is thicker and contains more elastic fibers in the arteries than in the veins, and is essential for maintaining blood pressure. Vascular smooth muscle cells (SMCs) have contractile phenotypes that engage in vasoconstriction or dilatation of blood vessels. The fiber composition is also different.

### 2.3. Tunica Externa

The outermost layer is an outer membrane composed entirely of bonded fibers. It consists of fibroblasts embedded in a loose collagen matrix. The outer membrane consists mainly of connective tissues such as type I and type II collagen, preventing blood vessels from stretching or shrinking excessively. These collagen fibers provide the strength of blood vessels or resistance to dilation due to excessive blood pressure (Figure 2). 

## 3. Materials Used in TEVG Design

An essential factor to consider in TEVGs is the selection of biomaterials associated with forming vascular tissue. Due to the complexity and variety of the vascular wall structure, the choice of biomaterials is not apparent and requires careful consideration and requirements. In vascular tissue engineering, three classes of materials are mainly used: synthetic polymers, natural polymers, and bio-derived materials such as a decellularized matrix. Since the possibility of inducing foreign body reactions cannot be denied, the view has changed to respond to the inflammatory response, the starting point of tissue regeneration. Furthermore, as a more realistic approach considering clinical applications, good handling of things such as blood leakage while applying small sutures, threading strength, ease of sterilization and storage, and versatility are also important. In addition to commonly used materials such as biodegradable polymers and biogenic materials, we will focus on the combination with cells in the regenerative vascular approach, classify them, and characterize them. In synthetic macromolecules, (1) synthetic polymers are often used in blood vessels and complex tissues such as bones. In addition, (2) fibrous proteins such as collagen and elastin constituting ECM and non-native natural polymers such as silk fibroin and chitosan with high biocompatibility have been widely studied as basic materials for single or complex use. Furthermore, (3) the use of various decellularized tissues has been carried out due to the high tissue affinity, and (4) in recent years, the development of composite materials that achieve this has flourished (Table 1).

### 3.1. Degradable Synthetic Polymers

A balance between the speed of material degradation and the rate of tissue formation is essential for the graft to maintain mechanical integrity and development of appropriate tissue remodeling. This challenge is best encompassed for arterial TEVGs placed in a high-pressure environment. The scaffold must resist arterial pressures at implantation and support adequate neotissue formation. The resulting neovessel may resist aneurysmal dilation or rupture through complete polymer degradation.

The most commonly studied and used synthetic polymers are polyglycolic acid (PGA) and polylactic acid (PLA) and their copolymers. Because they are FDA approved for use in humans, they have previously been used for resorbable sutures. PGA is flexible and has no inflammatory response. PGA can withstand mechanical stress equivalent to aortic pressure. Still, with a short degradation time of 6 to 8 weeks, it must be combined with other polymers for clinical application to blood vessels [22]. PLA is a polymer with structure and mechanical properties similar to those of PGA. Still, it has a longer decomposition time and can maintain tensile strength for more than a year in sutures [10]. However, PLA possesses a hydrophobic structure, making it unsuitable [22]. Therefore, poly(L-lactide) (PLLA), in which only L-lactic acid is polymerized, has been the most studied polymer in cardiovascular tissue engineering applications [23]. Each of these tends to have a shorter decomposition periods than for their use alone, allowing the mechanical properties to be adjusted depending on the composition ratio. Since PLLA has high biocompatibility and is relatively slow to decompose [24], sometimes it is used in combination with PGA. PCL is another biodegradable polyester that has advantages over PGA and PLA because Polycaprolactone (PCL) exhibits stronger maximum stress and tensile strength than natural blood vessels. PCL has excellent biocompatibility and slow biodegradability [13]. However, because PCL is hydrophobic, it is commonly combined with other polymers, rather than being used as a single base, to improve cell response, antithrombotic function, and tissue infiltration. Replacing a vascular graft of PCL nanofibers with a rat abdominal aorta, endothelialization and cell infiltration into the graft developed rapidly up to six months and maintained patency until 18 months. However, it has been pointed out that the cellular regression observed at 12 and 18 months may lead to calcification through intimal hyperplasia in the medium term [14]. The mechanical strength of the material and the rate of endothelialization, but also the need for degradation and cell infiltration along with remodeling, were demonstrated.

PU is often studied in combination with bio-derived materials described later, because PU reflects the design of the structure of the crystal domain based on chemical formulas and it is easy to control its biodegradability and mechanical properties. [12,16].

### 3.2. Natural Polymers

#### 3.2.1. Collagen, Elastin

Collagen, a major component of ECM, has been studied not only as cardiovascular but also as a tissue engineering material such as skin [25]. It has low antigenicity and good biocompatibility, promoting cell adhesion and proliferation. Elastin is also used because it has a cell adhesion site and helps impart flexibility, maintaining the elasticity of blood vessels under blood pressure [26]. In addition, due to the effect of smooth muscles on the phenotype, this material prevents the endometrial growth of natural blood vessels and provides the organization of collagen fibers. Focusing on the constituents and roles of the vessel wall, it has been reported that the elastin network contributes to the mechanical properties of the arterial wall [27]. In addition to their essential role in mediating ECM signaling, elastin and collagen are also involved in the elasticity and stiffness of maintaining pulse pressure and the operational control of SMCs function. This showed the effects of the density and amount of elastin in the three-dimensional porous structure on the graft with rigid arteries. Furthermore, it has been reported that when vascular smooth muscle cells are cultured in the form of elastin fibers, they show a contractile type and stop proliferation, as observed in native ECM [2]. Therefore, it has been shown that it is possible to reproduce both the mechanical and biological properties of natural elastin, but long-term evaluation in vivo is required.

#### 3.2.2. Silk Fibroin, Chitosan

A relatively frequently used biogenic material is biocompatible, biodegradable silk fibroin [28,29,30]. An insect called silkworm mainly produces this; this natural protein does not show immunogenicity in humans, promotes angiogenesis and tissue neogenesis, and exhibits good cell compatibility. Furthermore, because it is a crystalline protein with excellent thermal stability, it can be applied to various forms such as nonwoven fabrics, sponges, and coatings. It is possible to control decomposition engines, so it is attracting attention as a primary material. A cell-free two-layer material sponge-coated on a graft knitted with silk fibroin was used to perform bilateral end-to-end common carotid arteries (CCA) bypasses in a canine model for use in the coronary arteries or lower extremity arteries [18]. The sponge layer of the graft, open even a year later, had almost decomposed and been replaced with fibrous tissue. Furthermore, although a J-shaped curve similar to that of natural arteries was observed in vitro, its compliance was higher than that of ePTFE, but considerably lower than that of natural arteries. After four months of operation, the graft’s perfusion defect-free enhancement was confirmed, and the in vivo strength acquisition by remodeling was established.

Chitosan is a relatively new material used in TEVGs and has attracted attention due to its extremely low antigenicity, hemostasis, and antimicrobial effects on bacteria and fungi [31,32]. It is often used in combination with other materials, and the difficulty of processing as a material and inconvenience due to heterogeneity are being well improved.

### 3.3. Decellularized Tissue

Research into animal tissues for artificial blood vessels was started ahead of other materials. Strategies have been developed to remove immunogenic cellular components from human and other animal ECM tissues by physical, chemical, and enzymatic treatment, as well as combinations of these [33,34]. These matrices maintain the biological properties of natural blood vessels, and research has been advanced mainly in Europe because they are relatively accessible from organisms such as pigs and cattle. Still, these are inferior to synthetic materials due to the high risk of thrombosis, infections, and aneurysms, even at greater diameters of the size available [35].

In the first study of arterial decellularized tissue, the carotid artery of pigs was utilized [19]. The mechanical properties based on the elastin and collagen components showed a slightly lower maximum burst pressure than native tissue. In contrast, a significant decrease in the maximum stress was confirmed, indicating the problem of damage caused by treatment. There is a hybrid type of nanofiber for endometrial thickening by electrospinning of PCL in the decellularized rat aorta. The use of rapamycin release from the nanofiber layer enhanced mechanical properties avoiding rupture due to thrombus formation and suppressed endometrial growth. Although it is an example of a small diameter, research on very long vascular materials has been reported in recent years. Since the patency rate decreases as the vascular graft increases in size, blood vessels of the ostrich carotid artery modified with REDV peptide have been reported to induce reendothelialization [13]. Despite being 20 to 30 cm long, inhibition of thrombus formation was fully observed in the pig femoral artery after 20 months. In recent years, research producing material surfaces of decellularized tissue towards the cellular level has also been carried out [36]. A new antithrombotic additive material was reported for decellularizing ECs after culturing ECs in vitro on micropattern scaffolding.

### 3.4. Composite Materials

By processing biodegradable polymers into porous bodies such as nonwoven fabrics and sponges, customized cell-permeable graft designs are enabled for the desired blood vessels, that can withstand a variety of pressures, twists, and elongations. Although the mechanical properties following the pulsation and contraction of native tissue are very important, they are not suitable for cell adhesion and proliferation in a single use. On the other hand, natural materials have cell adhesion capabilities, but their mechanical properties are inferior. Nevertheless, both reendothelialization and mechanical properties are essential to the long-term portability of TEVGs [37]. For these reasons, hybrid grafts, with respect to aspects such as materials and fabrication, multilayer structures, and molding into 3D, have emerged. A different approach is the development of vascular structures using bioactive substances such as collagen seeded with vascular cells [8]. The extracellular matrices secreted by cells were used and associated with vascular cells to reconstruct blood vessels restored to permeability and biocompatibility [38]. Therefore, the ideal balance between proper mechanical strength and preferential use of natural biomaterials remains a challenge.

In addition, just as thrombus formation is an essential complication of TEVGs, it is necessary to consider the lining that comes into contact with the blood of blood vessels in particular. Therefore, many studies on the appropriateness of synthetic and biomaterials as the base material have also been carried out. Surface chemical modifications such as functional molecules as growth factors interact with cells [39], and are antithrombotic due to plasma treatment and water control for water wettability and protein adsorption modification [4,40,41,42]. Similarly, to induce endothelialization of the material surface early after transplantation, many approaches aim for antithrombotic by post modal modification of functional molecules to the material surface and blending into the material [43].

When a nonwoven fabric made by mixing silk fibroin and PU with SVVYGLR peptides with angiogenesis ability was buried in the rat abdominal aorta, many minute neovascular vessels were confirmed, in addition to the decomposition of materials and the formation of endothelial and smooth muscle layers, in which tissue infiltration and vascular neogenesis were similar to native blood vessels. These strategies affect the immediate effect and the longer-term performance [44,45].

### 3.5. 3D Bioprinting

3D bioprinting provides a solution by building specific vascular grafts with layered biomimetic structures. It manufactures vascular structures directly using different cell types [46]. Cell-type spatial control and the ability to print forms in layers are the advantages of 3D bioprinting over other metho. Therefore, 3D bioprinting is an engaging technology with the potential to produce patient-specific grafts. Currently, among the different categories of bioprinting methods, extruded bioprinting technology, in particular, is being widely explored in the field of vascular tissue engineering due to its excellent mechanical properties compared to other methods. In this section, we will introduce the materials used in bioprinting. 

Fibrin is a protein in the blood and plays an important role in blood clotting [47]. Fibrin provides excellent biocompatibility and degradation properties, but its mechanical properties are fragile. Fibrin has been used as a bio-ink for printing microchannels using human EC [48]. Drop-on-demand bioprinting techniques create a three-layer vascular model [49]. This study loaded human umbilical cord vein endothelial cells (HUVECs) into gelatin and printed them as sacrificial materials. The next layer containing SMCs in fibrinogen was printed to mimic the membrane of natural blood vessels. The stiffness of these printed samples was close to the rigidity of natural blood vessels. A composite bio-ink administered with gelatin and a blend of fibrinogen and gelatin was used to make vascularization using rotary bioprinting techniques [50]. Agarose is a natural polysaccharide that is also used in vitro experiments. However, it lacks the function of cell adhesion and proliferation. On the other hand, it has excellent biocompatibility and mechanical properties [51]. Agarose hydrogels were used for 3D bioprinting to support the cell aggregate to form linear and branched blood vessels [52]. Agarose is blended with collagen, alginate, and fibrin and has been reported to improve cell adhesions. Alginate is commonly used in the bioprinting of angiostructure. Alginic acid bio-ink was used in the inkjet bioprinting process [53]. However, the disadvantages associated with these bio-inks are their low degradability in vivo and insufficient cell adhesion and migration. Oxide alginate bio-inks were developed to solve these problems. In addition to the benefits of alginate hydrogels, oxide alginates provide faster degradation [54]. Collagen is one of the most important proteins in mammalian ECMs, making collagen a biomaterial for 3D bioprinting applications. Collagen has low fidelity in the printed shape and uneven cell distribution [55]. However, collagen bio-inks exhibit excellent cell adhesion and proliferation properties. Therefore, collagen bio-ink can be used with other biomaterials [56]. An improved material made from collagen is gelatin. Gelatin hydrogels are biomaterials with thermal gelling properties and excellent biocompatibility and degradability. However, gelatin hydrogels change to a solution for bio-ink above 37 °C [57]. In addition, gelatin is combined with other biomaterials to prevent pyrolysis. MSC hydrogels feature a high level of thermal stability without compromising the salient features of gelatin, such as biocompatibility and degradability. A double-layer vascular graft was printed using a complex bio-ink containing Gelatin methacryloyl (GelMA), hyaluronic acid, glycerol, and gelatin [58]. In addition, gelatin is combined with other biomaterials to prevent pyrolysis.

However, despite recent developments, there are no reports that bioprinted vascular grafts have yet been found to be effective in clinical trials, mainly due to challenges related to bio-inks. Although both natural and synthetic hydrogels have been tested, bio-inks provide an appropriate environment for cells and have sufficient mechanical strength. Bioprinted grafts made from hydrogels are fragile and have insufficient strength to withstand hemodynamic pressures. Further research is necessary to design synthetic bio-inks that can provide a bioactive, cell-friendly environment and their inherent adjustable mechanical behavior.

## 4. Cellular Factors

It is necessary to consider what kind of cells should be used to design and manufacture functional TEVGs. The endometrium serves as an anti-inflammatory and antithrombosis formation, especially within the walls of blood vessels. The layers within the vessel wall are mainly composed of SMCs and ECs, respectively. It is thought that practical antithrombotic function, immune function, and regenerative ability cannot be adversely serrated without cells seeding the layer inside the blood vessel wall. Therefore, it is common to perform cell seeding on TEVGs. The origin of cells is an important aspect to consider in TEVG transplantation, given that allogeneic cells cause graft rejection. Therefore, autologous cells are regarded as the preferred source of grafts. Separation of autologous vascular cells requires invasive procedures, and mesenchymal stem cells (MSCs) and induced pluripotent stem cells (iPSCs) are currently being studied to solve the problem of adult cell proliferation ability (Figure 3) [59,60,61,62].

### 4.1. Autologous Primary Cells

Patient-derived autologous cells are a potential source of interest because they may minimize graft rejection. The 50:50 copolymer of the PLA and PCL, designed to undergo several months of hydrolysis in vivo, was combined with a PLA or PGA nonwoven fabric to reinforce the structure [63,64]. When canine femoral vein-derived myofibroblasts were buried as autologous vascular grafts in the inferior vena cava of the canine model after seeding, the formation of the endothelial layer and the production of elastic fibers were confirmed in six months. In addition, structures similar to natural blood vessels that do not form thrombus or blockage were obtained. The effectiveness of the combination of biodegradable scaffolding and autologous cell seeding in the early stages of TEVGs development was shown. However, separating and culturing viable and eligible primary cells to a therapeutically appropriate scale is likely a bottleneck. An essential factor to consider first is cell proliferation, which decreases with the age of the donor, but also depends on the cell type selected [65]. Due to donor site morbidity and risk of contamination so, autologous stem cells are a more practical alternative.

### 4.2. Autologous Stem Cells

MSCs are multipotent cells that can be cultured in skeletal muscles, blood vessels, and bones [36]. Research is being actively conducted, because it is possible to efficiently separate them from bone marrow and culture. The immunomodulatory properties of MSCs are also attracting attention, because they act to modulate foreign-body reactions to materials in addition to tissue repair and regeneration [66]. Adult stem cells, such as MSCs, have the advantage of being usable immediately after separation. Still, their survival rate largely depends on the patient’s health and age, leading to complex treatment strategies and verifications. This is why embryonic stem cells (ESCs) are increasingly used in vascular tissue engineering research. However, when it comes to clinical use, it is necessary to solve the ethical and potential risks of tumor formation.

### 4.3. IPSCs

With the advancement of stem cell technology [67], autologous and iPSCs research is developing rapidly [68]. When smooth iPSC-derived vascular muscles were sown in PGA scaffolding and buried under the skin of immunodeficiency mice, vascular tissue development and maturation were confirmed [69]. Finally, iPSC has the advantage that the number of cells available is almost unlimited and can differentiate into vascular-specific strains. However, all cell sources have their limitations, and like ESC, iPSC raises concerns about the fate of transplanted cells and their possible role in tumorigenesis [70,71]. In addition, each laboratory has its differentiation protocol. Before these differentiated cells can be used in clinical applications, there are institutional challenges to establishing reliable and efficient protocols that show the effect of each differentiation step on the function of the obtained cells.

## 5. TEVG Remodeling and In Vivo Integration (Pre-Clinical and Clinical Section)

The goal of vascular tissue engineering is to recapitulate the structure and function of the native vessel. The success of vascular remodeling depends on two main components. One is polymer degradation, and the other is neotissue formation. Depending on the graft polymer type, degradation speed is different. On the other hand, the graft pore size and fiber lining cell infiltration can influence neotissue formation. In a perfect world, as a polymer degrades, neotissue should start forming and laying down necessary cell types to maintain the vascular hemodynamic pressure. Unfortunately, in reality, animal and human bodies recognize TEVGs as foreign objects and start sending immune cells to attack them. As it sends a stronger signal to recruit more cells to attack it, it forms thicker scar tissue, which forms stenosis. Although there are many methods for halting stenosis, if one is not careful with this process, it could stop the cell infiltration of cells that produce extracellular matrix, which leads to aneurysmal change. Fine tuning of each component is the key to getting closer to a functioning native vessel.

### 5.1. Pre-Clinical Models

Humphrey and Breuer et al. worked together for a decade to combine large animal model surgical experiments and computer biomechanical simulation to further understand and optimize TEVG remodeling for perfect TEVGs for clinical use. Their main work describes the inflammatory responses to different graft material structures [72].

For example, two-layer grafts with silk mesh and silk sponge have been developed to withstand long-term pressure by mimicking the layer structure of native blood vessels. It was implanted in the femoral arteries of dogs for one year. At three months after implantation, the development of the vascular endothelial layer on the surface of the lumen was confirmed. In addition, the result of collagen and elastin fibers over time on the luminal surface of blood vessels has also been established, which may affect long-term vascular occlusion and graft deformation [72].

### 5.2. Clinical Models

Long-term follow-up in large animals is an important step (Figure 4). However long-term comprehensive assessment of the graft performance, beyond complete scaffold degradation, should be performed in humans because, especially in the pediatric field, we need TEVG, which grows with the patient’s growth. Based on animal experiments using the bone marrow mononuclear cells (BM-MNC) seed scaffold, Shinoka et al. started a clinical trial to evaluate the use of TEVG and patches for congenital heart surgery [73]. Twenty-three patients had a tube graft as an extracardiac conduit-total cavopulmonary connection (EC-TCPC). In comparison, the other 19 patients had a sheet-type patch used to repair congenital cardiac defects. In contrast with PET and ePTFE graft recipients who typically remain on long-term anticoagulation therapy, 96% of patients were able to discontinue anticoagulation therapy six months after the TEVG operation [74,75].

There have been very few clinical cases; however, there are cases in which autologous peripheral venous cells were seeded in a single graft in the early stages. Shinoka et al. transplanted cells from their self-peripheral veins to TEVGs, which consisted of PLA: PCL = 50:50 copolymer. TEVG, including the obtained autologous vascular cells, reported a successful transplant of TEVG, replacing a 2 cm portion of the pulmonary artery in a 4-year-old girl. The biodegradable polymer TEVG was 10 mm in diameter, 20 mm long, and 1 mm thick, and was designed to decompose over 8 weeks. As the cells increased over eight weeks, the material was designed to decompose according to the cells increased. There were no postoperative complications, and follow-up angiography showed that the graft was utterly patent. This is the result obtained by seeding cells derived from the patient, and effective results in Fontan surgery have been obtained with large sizes of this as well [76].

Shinoka et al. previously conducted clinical trials in vein models. TEVGs with a combination of materials on cell-free grafts were developed and tested, but the cell seeding did not match the non-cell seeding, where they narrowed and developed clots inside the vascular grafts. Therefore, studies have focused on developing grafts without sowing the cell on the graft. However, there are still some challenges in using non-cell TEVGs in clinical applications [77].

## 6. Challenges and Future Prospects

While autologous tissue has successfully been applied to treat many cardiovascular diseases (CVDs) (e.g., atherosclerosis), autologous vascular tissue for transplantation is often insufficient [12]. Therefore, the development of tissue-engineered vascular grafts (TEVGs) aims to provide alternatives to autologous tissue. Although many studies have been published using different materials and methods for developing TEVGs, some of the limitations associated with conventional grafts remain, such as the occurrence of blood clots. In this work, we reviewed the results recently published in the field of TEVG development for use in both animal models and humans.

Advances in TEVG manufacturing technology may soon be able to address these issues. Additionally, it is now possible to 3D print vascular grafts with satisfactory properties. To maintain long-term graft patency, the mechanical properties and cell mobilization of TEVGs must be controlled. However, advances in these technologies still do not allow the generation of vascular tissue according to the demands of all patients. The development of hybrid grafts, combined with natural materials and biodegradable polymers, now essentially emulates the original blood vessels, so safe clinical use of TEVGs in human patients is approaching. Future research that leverages continuous innovation and discovery in vascular and biological engineering, biomaterials, and stem cell engineering will create constant motivation and hope for translatable strategies.

## Figures and Tables

**Figure 1 biomedicines-10-01439-f001:**
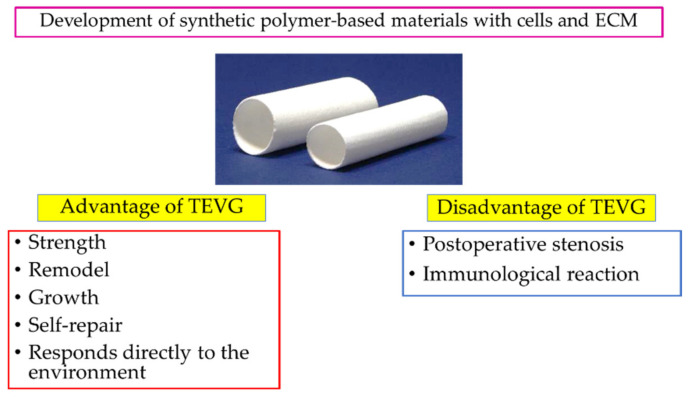
The advantages and disadvantages of TEVGs have emerged over years of development. Recently, the focus has been on the development of next-generation TEVGs with cells and ECM in synthetic polymer-based materials. The image was produced by the authors using the photos provided by Gunze Ltd. (Tokyo, Japan).

**Figure 2 biomedicines-10-01439-f002:**
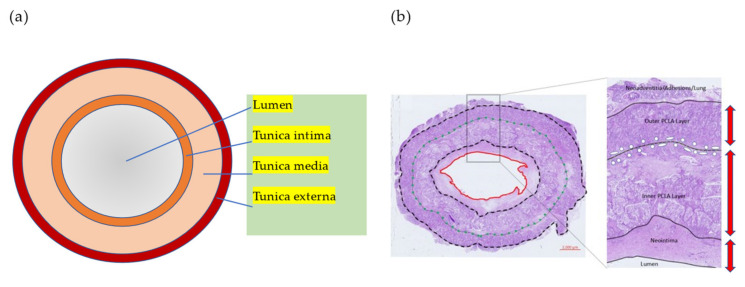
(**a**) Typical structure diagram of blood vessels. Blood vessels consist of three layers: tunica intima, tunica media, and tunica externa. (**b**) Tissue staining image of a typical TEVG fabricated with polylactide-co-caprolactone (PCLA) and polyglycolic acid (PGA) in sheep models. After only six weeks, TEVG has remodeled three layers like a native vessel structure. The image was produced by the authors using images from our previous study.

**Figure 3 biomedicines-10-01439-f003:**
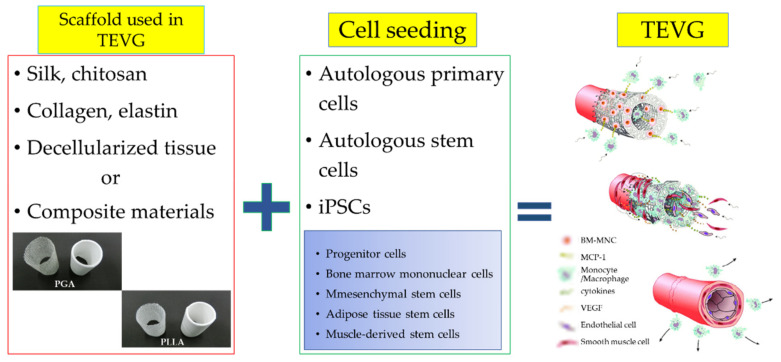
Progressive remodeling in TEVGs in which cells were seeded on scaffold material. It was produced by the authors using the photos provided by Gunze Ltd. (Tokyo, Japan).

**Figure 4 biomedicines-10-01439-f004:**
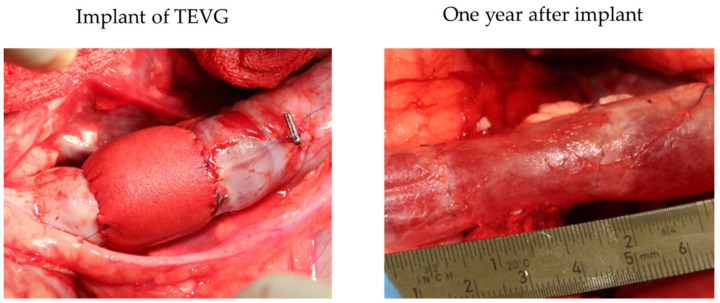
This figure shows immediately after the TEVG implantation (**left**) and one year after the TEVG implant (**right**) in sheep models. As shown in the figure, one year after implanting the TEVG, it had adapted to the native blood vessel. The figure was produced by the authors using the images from our previous study.

**Table 1 biomedicines-10-01439-t001:** Studies of vascular grafts for achieving good mechanical properties and patency.

Manufacturing Method	Component	Development Level	Comments	Refs
Gore-tex^®^	ePTFE	in vitro in vivo (pig model)	Low patency; only two/seven (29%) after six months. Mechanical properties deviate significantly from living tissue. Compliance of 0.0034 ± 0.0004 (%/mmHg) from 80 to 120 mmHg.	[14,15]
electrospinning fiber	PCL	in vivo (rat model)	Patency until 18 months. Reendothelialization and cell infiltration into the graft developed rapidly for up to six months.	[14]
electrospinning fiber	PU/PCL PU/PCL-heparin	in vitro in vivo (rabbit model)	A dual functional polyurethane for mimics of blood vessel inner surfaces by combining surface texture and nitric oxide (NO) release. Compliance of 0.0360 ± 0.0018 (%/mmHg) from 80 to 120 mmHg. Patency after 5 months with increased intimal thickening and blood flow speed	[12,16]
Mesh coated in additional polymers	P(LA/CL) and PGA or PLLA, autologous BM-MNCs	in vivo (human trial)	First human clinical trial. Complete disassembly of scaffolding.	[17]
Mesh with a coating(sponge)	Silk fibroin/ silk fibroin	in vivo (canine model)	Patency until one year. Development of elastic fibers and progress of endothelialization. Compliance of 0.019 (%/mmHg) at 100 mmHg.	[18]
Decellularized tissue	Pig’s carotid artery	in vitro	First decellularized tissue material. Slightly lower maximum burst pressure than native tissue.	[19]
Decellularized tissue	Human placenta, Tissues crosslinked by riboflavin-mediated UV and coating with heparin	in vivo (rat model)	Patency after four weeks without the use of anticoagulants. Compliance of 0.094 (%/mmHg) at 100 (mmHg) before implanting. Rapid cell migration (host cells migrated from the lumen and the adventitial side into the vessel walls) and vascular graft remodeling decreased graft compliance.	[20]
Decellularized tissue with electrospinning fiber	Rat’s aorta, PCL blended with rapamycin	in vivo (rat model)	Sustained release of the drug from the PCL nanofiber layer reduces neointimal hyperplasia. Progression of reendothelialization and M2 macrophage polarization at 12 weeks.	[21]
Decellularized tissue	Ostrich carotid artery modified with REDV peptide	in vivo (pig model)	Long bypass graft 20–30 cm in length. No thrombus formation on the luminal surface during 20 days of observation without anticoagulant administration.	[13]

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
