# Peer review of "Clinical Application for Tissue Engineering Focused on Materials"

_biomedicines, 2022, doi:10.3390/biomedicines10061439_

Round 1
Reviewer 1 Report
Dear authors, congratulations on the excellent review article.
Review
In the present review article, the clinical possibilities of tissue engineered large lumen vascular grafts are presented. In particular, the selection of polymers as biomaterials and cell sources and the associated clinical trials are discussed. The introduction begins with a brief historical overview of vascular grafts made from synthetic materials and describes the need for TEVGs. In the second part of the review, the structural composition of a vessel is presented and then in part three materials, synthetic and biobased, that can be used and have been investigated in studies are presented. The cellular aspects are systematically presented briefly and concisely in part 4. Preclinical and clinical data are then discussed.
The article cites 63 scientific papers that give a general overview of the field.
Line 139: Figure 2: The caption of the figure is not readable. The abbreviations PCLA and PGA should be written out in the description of the figure. The origin of the figures should also be indicated. If it is the author's own work, this should be stated.
Line 180: L-lactide - hyphen
Line 183:
What is meant by coprolytic body?
PLLA is not a copolymer of lactic acid and caprolactone. The monomer is L-lactic acid, and thus poly(L-lactic acid). L denotes the L-enatiomer.
The copolymer of lactic acid and caprolactone is variously called PLLA-PCL, PLLA-co-PCL, PCL-co-PLLA or PLCL. In the literature cited [15] (Ong, C.S.; Fukunishi, et al.) the polyglycolic acid (PGA)/polylactide-co-caprolactone (PLCL) system is used. Pure PLLA does not occur here.
Please check this section again.
Lines 197 to 199
The literature cited [18] (Xu, L.C.; Meyerhoff, M.E.;et al.) deals with a commercially available PU (CarboSil® from DSM Biomedical Inc.) with a structured surface and NO release. It does not talk about controlling biodegradation and mechanical properties! Please check the lit. here.
Line 278
The cited literature [14] and [35] (I have not checked the other citations) in the table do not seem to be correct; perhaps check the literature citations again, something may have got mixed up.
Line 294: Native without capital letter
Line 308: Figure 3: The caption in the right part of the picture under TEVG is not readable; the caption should be self-explanatory.
Line 584: Adjust capital letters in Lit 59.
Reviewer 2 Report
Summary: In the current manuscript, Kitsuka et al. claim that they give an overview of the biomaterials and cell types as components of vascular tissue engineering and related clinical trials aiming at developing large diameter vascular grafts.
Hereinafter are shown my comments that the authors should consider:
1. Although I am not a native English speaker, I recommend that a proofreading in terms of the written English should be applied to the whole manuscript. Please see several examples: “The mechanical properties based on the erpsyin and collagen components showed a slightly lower maximum burst pressure than Native” (lines 248-249); “In recent years, research on the surface of materials by producing decellular tissue at the cellular level has also been carried out” (lines 259-260); “Therefore, many studies on the fine of synthetic and biomaterials as the base material are also carried out. Surface chemical modifications such as functional molecules as growth factors interact with cells [38] and are antithrombotic by plasma treatment and water control for water wettability and protein adsorption modification” (lines 282-286) etc. Besides, the authors used inappropriate constructions such as: “related clinical trials in developing large diameter vascular grafts that can be applied clinically”; “When iPSC-derived vascular smooth muscles were sown in scaffolding and buried under the skin…”; “decellular tissue”;
2.Title: I suggest the authors to rephrase the manuscript title since “Clinical Application for Tissue Engineering” is a general term as well as an inappropriate construction. Furthermore, it does not reflect exactly the content of the manuscript where including preclinical models and the components of vascular tissue engineering are presented. Besides, despite the great advance of the state of art in the field, the manuscript is not well-documented.
3. Keywords: “tissue engineered vascular grafts” and “TEVGs” are equivalent. So, please choose one of them.
4. Figures: They must contain a title and a legend underneath. The title is absent, and authors present rather some comments that could be shown as legend. Please revise.
5. Figure 5: Is the image corresponding to TEVG drawn by the authors? Please quote the paper.
6. Please expand the abbreviations at their first use within the manuscript, e.g., PCLA, BM-MNCs.
7. Author Contributions are not mentioned. Please revise.
8. The authors are advised to check and revise the References section. For instance, ref. [7] does not include all authors; ref. [21] is incomplete, etc.
Reviewer 3 Report
The review article entitled “Clinical Application for Tissue Engineering” features a systematic review of tissue-engineered vascular grafts (TEVGs) focusing on their importance in clinical application. The review also highlights the selection of biomaterials and cell sources as well as preparation methods. The manuscript lacks a few important topics such as 3D printing which is an important method for scaffold production as well as a few natural polymers like Elastin. Thus, issues are needed to be addressed first before the recommendation of this review article for publication
1. Line 31: What is CVD. Provide full-form at first use
2. Line 242: “immunogenic cellular components from 241 human and other animal tissues by physical and chemical treatment and ECM.” why ECM here and explain.
3. Add the note on the application of Dacron® (polyethene terephthalate (PET)) and Teflon® (expanded polytetrafluoroethylene (ePTFE)) in Tissue-engineered vascular grafts (TEVGs) preparation.
4. Elastin is an important natural polymer used in TEVGs graft. Add a section on Elastin.
5. 3D printing is an important method used for the production of the scaffold in tissue engineering. Add a section on ‘3D printing technologies utilized for the TEVGs graft.
6. Are MSCs is pluripotent stem cells in nature?? verify
7. The author should cite the recent important reference in this review article. such as https://doi.org/10.1016/j.bioactmat.2020.12.021, https://doi.org/10.1016/j.actbio.2021.06.034
8. Typographical errors need to be corrected throughout the manuscript.
Author Response
Please see the attachement.

Round 2
Reviewer 2 Report
The authors appropriately addressed my comments as
well as the concerns raised by the other reviewers
and the revised manuscript fulfils the standards to be considered
for publication in Biomedicine journal.